# Post-Transcriptional Induction of the Antiviral Host Factor GILT/IFI30 by Interferon Gamma

**DOI:** 10.3390/ijms25179663

**Published:** 2024-09-06

**Authors:** Taisuke Nakamura, Mai Izumida, Manya Bakatumana Hans, Shuichi Suzuki, Kensuke Takahashi, Hideki Hayashi, Koya Ariyoshi, Yoshinao Kubo

**Affiliations:** 1Department of Clinical Medicine, Institute of Tropical Medicine, Nagasaki University, Nagasaki 852-8523, Japan; nakamurat@nms.ac.jp (T.N.); mizumida@nagasaki-u.ac.jp (M.I.); manyabakat@gmail.com (M.B.H.); kari@nagasaki-u.ac.jp (K.A.); 2Program for Nurturing Global Leaders in Tropical Medicine and Emerging Communicable Diseases, Graduate School of Biomedical Sciences, Nagasaki University, Nagasaki 852-8523, Japan; 3School of Tropical Medicine and Global Health, Nagasaki University, Nagasaki 852-8523, Japan; suzuki_shuichi@nagasaki-u.ac.jp (S.S.); kensuket@nagasaki-u.ac.jp (K.T.); 4San Lazaro Hospital-Nagasaki University Collaborative Research Office, Manila 1003, Philippines; 5Medical University Research Administration, Nagasaki University School of Medicine, Nagasaki 852-8523, Japan; hhayashi@nagasaki-u.ac.jp

**Keywords:** GILT, post-transcriptional regulation, rapamycin, interferon gamma

## Abstract

Gamma-interferon-inducible lysosomal thiol reductase (GILT) plays pivotal roles in both adaptive and innate immunities. GILT exhibits constitutive expression within antigen-presenting cells, whereas in other cell types, its expression is induced by interferon gamma (IFN-γ). Gaining insights into the precise molecular mechanism governing the induction of GILT protein by IFN-γ is of paramount importance for adaptive and innate immunities. In this study, we found that the 5′ segment of GILT mRNA inhibited GILT protein expression regardless of the presence of IFN-γ. Conversely, the 3′ segment of GILT mRNA suppressed GILT protein expression in the absence of IFN-γ, but it loses this inhibitory effect in its presence. Although the mTOR inhibitor rapamycin suppressed the induction of GILT protein expression by IFN-γ, the expression from luciferase sequence containing the 3′ segment of GILT mRNA was resistant to rapamycin in the presence of IFN-γ, but not in its absence. Collectively, this study elucidates the mechanism behind GILT induction by IFN-γ: in the absence of IFN-γ, GILT mRNA is constitutively transcribed, but the translation process is hindered by both the 5′ and 3′ segments. Upon exposure to IFN-γ, a translation inhibitor bound to the 3′ segment is liberated, and a translation activator interacts with the 3′ segment to trigger the initiation of GILT translation.

## 1. Introduction

Gamma-interferon-inducible lysosomal thiol reductase (GILT), also known as interferon gamma inducible protein 30 (IFI30), serves as a multifunctional host protein with pivotal roles in both adaptive and innate immunities. Initially identified as a critical host factor, GILT facilitates the digestion of disulfide bonds present on antigen proteins within antigen-presenting cells including macrophages and dendritic cells. Notably, GILT deficiency in mice has been linked to impaired T cell responses against mouse herpes virus [1,2]. Following the internalization of antigen proteins into antigen-presenting cells through phagocytosis, GILT-mediated cleavage of disulfide bonds renders these antigens more susceptible to degradation by endosomal proteases including cathepsins [3,4]. The resulting peptides are subsequently presented onto major histocompatibility complexes, thereby activating T lymphocytes. Clearly, GILT plays a pivotal role in initiating adaptive immunity.

Nonetheless, GILT’s conservation across numerous invertebrates, which lack adaptive immunity, and its presence in plants and fungi underscore the potential existence of unknown functions for GILT beyond adaptive immunity [5]. Our previous investigations have revealed that GILT exerts direct inhibitory effects on diverse viral infections through the cleavage of disulfide bonds within viral envelope glycoproteins, implying an additional role for GILT as an innate immune factor against viruses [5,6]. Since this initial discovery, substantial evidence has accumulated from various independent research groups, further elucidating GILT’s involvement in innate immunity. For instance, the mosquito GILT homolog impedes *Plasmodium sporozoite* transmission [7], the fruit fly GILT homolog curtails bacterial growth [8], the shrimp GILT homolog restricts white spot syndrome virus infection in vivo [9], and human GILT suppresses infections caused by SARS-CoV-1, Ebola virus, and Lassa virus [10]. Collectively, GILT assumes diverse roles in both adaptive and innate immunities.

Hence, unraveling the molecular mechanisms governing GILT expression regulation promises novel insights into adaptive and innate immunities. While GILT is consistently expressed in professional antigen-presenting cells like macrophages and dendritic cells, its expression in other cell types is induced by interferon gamma (IFN-γ). In broad terms, the interaction between IFN-γ and its cell surface receptor triggers the activation of Jak 1 and 2. These activated Jak 1 and 2 enzymes phosphorylate the STAT1 protein, leading to the transportation of phosphorylated STAT1 to the nucleus. This nuclear translocation activates the transcription of target genes [11].

However, a microarray analysis of HeLa cells subjected to IFN-γ treatment disclosed that GILT mRNA expression remains relatively high in the absence of IFN-γ, with no substantial increase following IFN-γ treatment [12]. This intriguing result suggests either the stabilization of GILT protein in response to IFN-γ or the activation of GILT protein translation by IFN-γ. In light of these findings, our study aims to decipher the precise molecular mechanisms by which IFN-γ induces GILT protein expression.

## 2. Results

### 2.1. Post-Transcriptional Elevation of GILT Protein by IFN-γ in HeLa Cells

To investigate the influence of IFN-γ on GILT protein expression in HeLa cells, we introduced IFN-γ to the cells and prepared cell lysates at 0, 1, 2, and 3 days post-treatment. The GILT protein expression was analyzed via western blotting of the cell lysates. The GILT protein levels in cells treated with IFN-γ were higher than those in untreated cells, with a particularly marked increase observed at day 3 after IFN-γ exposure (Figure 1A).

However, our previous microarray analysis indicated comparable GILT mRNA levels in HeLa cells both untreated and treated with IFN-γ for 3 days (Figure 1B) [12]. In contrast, the mRNA levels of other IFN-γ-inducible host factors, FAT10, IDO1, and IFI6, were significantly elevated upon IFN-γ treatment, validating the activation of the IFN-γ signal transduction pathway in the treated cells. Notably, the GILT mRNA level remained unaltered.

To confirm the result, HeLa cells were exposed to IFN-γ for 3 days, and, subsequently, total RNA samples were prepared for reverse transcription. The copy numbers of GAPDH, GILT, and IFI6 cDNAs were quantified using droplet digital PCR (ddPCR). The normalized GILT and IFI6 copy numbers are indicated in Figure 1C. In line with the microarray result, the copy numbers of GILT remained unchanged following IFN-γ treatment, while those of IFI6 showed a significant increase. These collective findings emphasize that IFN-γ does not activate GILT transcription.

Transcription and translation occur in the nuclei and cytoplasm, respectively. Therefore, the transport of GILT mRNA from the nuclei to the cytoplasm may be inhibited in untreated HeLa cells to suppress GILT protein expression. To test this hypothesis, RNA samples were isolated from the cytoplasmic and nuclear fractions of untreated and IFN-γ-treated HeLa cells. The copy numbers of GILT and GAPDH cDNAs were quantified using ddPCR, and the normalized copy numbers of GILT cDNA in the cytoplasm and nuclei were compared. The localization of GILT mRNA was unaffected by IFN-γ treatment (Figure 1D). These findings suggest that GILT protein expression in untreated cells is inhibited by a mechanism other than the suppression of GILT mRNA transport to the cytoplasm.

### 2.2. Post-Transcriptional Elevation of GILT Protein by IFN-γ in Other Cell Types

To investigate whether the post-transcriptional upregulation of GILT protein expression occurs in other cells, we analyzed the IFN-γ response in human TE671 rhabdomyosarcoma and JEG3 choriocarcinoma cell lines, which have been maintained in our laboratory for a long time, as well as human peripheral blood mononuclear cells (PBMCs) isolated from a healthy donor. These cells were treated with IFN-γ for a duration of 3 days, and cell lysates and total RNA samples were then extracted from the treated cells.

Western blot analysis revealed an elevated GILT protein level in TE671 cells following IFN-γ treatment, while JEG3 cells did not exhibit an increase in GILT protein expression even in the presence of IFN-γ (Figure 2A). However, phosphorylated STAT1 protein clearly appeared in the IFN-γ-treated JEG3 cells like in HeLa and TE671 cells (Figure 2B), and ddPCR analysis showed that the level of IFI6 mRNA was dramatically elevated in the IFN-γ-treated TE671 and JEG3 cells (Figure 2C), indicating that the treatment activates the typical IFN-γ signal transduction in JEG3 cells. Untreated PBMCs displayed a relatively high level of GILT protein, and IFN-γ treatment did not result in a significant increase in its protein expression. These results indicate that TE671 cells elevate the expression level of GILT protein in response to IFN-γ.

To assess whether IFN-γ treatment upregulates GILT mRNA levels in TE671 cells, we measured the copy numbers of GAPDH, GILT, and IFI6 mRNA using ddPCR with the total RNA samples isolated from untreated and IFN-γ-treated TE671 cells. Copy numbers of GILT and IFI6 were normalized to those of GAPDH. The result showed that the normalized copy number of GILT mRNA remained unchanged following IFN-γ treatment (Figure 2C). However, the normalized copy number of IFI6 mRNA increased by two times, indicating the activation of IFN-γ signaling in the IFN-γ-treated TE671 cells. These results reveal that the induction of GILT protein expression by γ-IFN occurs post-transcriptionally in TE671 cells, mirroring the observed pattern in HeLa cells.

### 2.3. Absence of Activation of GILT Transcription by IFN-γ

To further validate our conclusion regarding the absence of activation of GILT transcription by IFN-γ, we performed PCR amplification of the GILT promoter region from genomic DNA isolated from HeLa cells. This region encompasses three potential IFN-γ-activating sequences (GASs) (Figure 3A). Subsequently, we fused the GILT promoter sequence with the nano luciferase (NanoLuc)-coding sequence derived from *Oplophorus gracilirostris* (GILT-NanoLuc) [13]. Additionally, a firefly luciferase expression plasmid under the control of GAS from the LMP2 promoter (GAS-FLuc) [14] was used as a positive control. Upon transfection of HeLa cells with the GILT-NanoLuc expression plasmid, the cells were treated with either PBS or IFN-γ 24 h post-transfection. The NanoLuc activities were quantified in cell lysates obtained 3 days post-treatment. The NanoLuc activity exhibited minimal variation in IFN-γ-treated versus untreated HeLa cells (Figure 3B). Conversely, the activity of firefly luciferase in HeLa cells transfected with the GAS-FLuc was significantly elevated by IFN-γ, indicating effective activation of the IFN-γ signal pathway. These findings solidify our conclusion that GILT protein induction by IFN-γ is not facilitated through activation of the GILT promoter.

### 2.4. GILT Protein Stability and Its Relationship with IFN-γ

In exploring the post-transcriptional mechanisms underlying GILT protein induction by IFN-γ, we examined two possibilities: enhanced protein stability and activated translation. We assessed the stability of the GILT protein under a condition with or without IFN-γ. HeLa cells stably expressing GILT protein were constructed using an MLV vector encoding the GILT protein without both the 5′ and 3′ UTRs of GILT mRNA (see Section 4). These cells were treated with a translation inhibitor (cycloheximide). The time course of GILT protein levels in the absence or presence of IFN-γ was analyzed with western blotting using an anti-GILT antibody. Although the cycloheximide treatment led to a gradual decrease in GILT protein levels over time, there was little difference observed between cell lysates with and without IFN-γ (Figure 4A,B). Collectively, these observations indicate that GILT protein stability remains unaltered by IFN-γ, further suggesting that IFN-γ likely enhances GILT translation.

### 2.5. Impact of Rapamycin on GILT Protein Induction by IFN-γ

A previous report demonstrated that an mTOR inhibitor, rapamycin, suppresses the expression of another antiviral host factor IFITM3 in HeLa cells [15]. In light of this finding, we investigated whether GILT protein induction by IFN-γ similarly relies on mTOR. To elucidate this, HeLa cells were concurrently treated with IFN-γ and rapamycin, and subsequent analysis of GILT protein levels was performed via western blotting. Interestingly, GILT protein was detected in HeLa cells treated solely with IFN-γ, but not in those treated simultaneously with IFN-γ and rapamycin (Figure 5A). Rapamycin treatment had no adverse effects on cell viability or growth (Figure 5B).

We further examined the impact of rapamycin on GILT promoter activity and IFN-γ signaling. Transfection of HeLa cells with GILT-NanoLuc plasmid, followed by treatment with IFN-γ and/or rapamycin, showed that rapamycin did not alter GILT promoter activity in response to IFN-γ (Figure 5C, left panel). Additionally, the examination of IFN-γ signaling using the GAS-FLuc expression plasmid confirmed that rapamycin did not interfere with the activation of GAS promoter by IFN-γ (Figure 5C, right panel).

To validate the result, HeLa cells were subjected to a 3-day treatment with IFN-γ and/or rapamycin. Total RNA samples were then prepared and subjected to the following reverse transcription to determine the copy numbers of GILT mRNA with ddPCR. The GILT copy numbers were unaltered following rapamycin treatment irrespective of the presence of IFN-γ (Figure 5D, left panel). The copy numbers of IFI6 mRNA were significantly elevated when cells were treated with IFN-γ, but the rapamycin treatment had no effect (Figure 5D, right panel). These results strongly support the conclusion that rapamycin attenuated the induction of GILT protein expression by IFN-γ through inactivating translation rather than through suppressing the transcription.

### 2.6. Untranslated Region of GILT mRNA Inhibits Its Translation in the Absence of IFN-γ

The mRNA levels of GILT in the absence of IFN-γ were relatively higher, whereas IFN-γ activates the translation of GILT protein. Consequently, it is possible that translation from the GILT mRNA could be hindered in the absence of IFN-γ. Notably, when HeLa cells were transduced with an MLV vector containing the GILT protein-coding region without the 5′ and 3′ untranslated regions (UTRs), these cells efficiently expressed GILT protein even without IFN-γ (Figure 4A). This suggests that the UTRs of the GILT mRNA might be responsible for inhibiting GILT protein translation.

To investigate this possibility, we acquired other GILT expression plasmids containing its UTRs. A fragment consisting of the 5′ UTR and the GILT protein-coding region was amplified using PCR and then ligated to the pcDNA3.1 expression plasmid, forming the 5′UTR-GILT construct (Figure 6A). Another fragment containing the GILT protein-coding region and the 3′ UTR was also amplified, forming the GILT-3′UTR construct. These plasmids were introduced into HeLa cells together with a luciferase expression plasmid in which the expression was under the control of MLV LTR, followed by treatment with IFN-γ for 1 day because endogenous GILT protein was not elevated by IFN-γ treatment for 1 day (Figure 1A). Cell lysates were prepared from the treated cells. GILT protein levels were assessed via western blotting, and their luciferase activities were measured to estimate the transfection efficiency.

In the absence of IFN-γ, GILT protein was not detected in HeLa cells transfected with either the 5′UTR-GILT or GILT-3′UTR constructs (Figure 6B), indicating inhibitory roles for both the 5′ and 3′ UTRs in GILT translation. However, upon the treatment of HeLa cells transfected with the GILT-3′UTR construct with IFN-γ, GILT protein expression was evident, although the luciferase activity in those cells was comparable to the luciferase activity of other cells (Figure 6C). This finding supports the notion that the 3′ UTR of the GILT mRNA contains a sequence that promotes GILT protein translation in response to IFN-γ.

To further validate this result, the 5′ and 3′ UTR sequences of the GILT mRNA were amplified using PCR and subsequently fused with the Renilla luciferase (RLuc)-coding sequence, as depicted in Figure 7A. Equal amounts of these expression plasmids were transfected into HeLa cells. Contrary to expectations, similar levels of luminescence were observed in the transfected cells (Figure 7B), suggesting that the sole UTRs do not have an impact on RLuc translation.

Hence, the 5′ or 3′ segments of the full-length GILT cDNA containing the full UTR and a part of the protein-coding region were combined into the RLuc coding sequence, as indicated in Figure 7C. The RLuc expression plasmids containing the 5′ and 3′ segments are designated as CMV-5′GILT-RLuc and CMV-RLuc-3′GILT, respectively. Equal amounts of the indicated expression plasmids were transfected into HeLa cells, followed by IFN-γ treatment for 2 days. Subsequently, RLuc activity was measured in the transfected cells. The presence of the translation start codon in the 5′ segment of GILT cDNA can hinder translation from the RLuc start codon. This impediment occurs, even if the 5′ segment lacks cis-acting inhibitory sequences against translation. Indeed, luminescence levels in the CMV-5′GILT-RLuc-transfected cells were much lower than those in the CMV-RLuc- or CMV-RLuc-3′GILT-transfected cells (Figure 7D). Conversely, the CMV-RLuc-3′GILT construct lacks additional sequences upstream of the RLuc. Notably, when the transfected cells were untreated, the RLuc activity of CMV-RLuc-3′GILT-transfected cells was lower than that of CMV-RLuc-transfected cells, suggesting the presence of a translation inhibitory effect within the 3′ segment of the GILT mRNA in the absence of IFN-γ.

In the presence of IFN-γ, the relative luminescence observed in CMV-5′GILT-RLuc-transfected cells remained lower than that in CMV-RLuc-transfected cells. This observation supports the conclusion that the 5′ segment of the GILT mRNA exerts a translation inhibitory effect, even when IFN-γ is present. However, the relative luminescence in CMV-5′GILT-RLuc-transfected cells was increased in the presence of IFN-γ compared to its absence, indicating a reduced translation inhibitory effect of the 5′ segment in the presence of IFN-γ.

The relative luminescence of cells transfected with CMV-RLuc-3′GILT was similar to those transfected with CMV-RLuc when the transfected cells were treated with IFN-γ, indicating efficient RLuc protein expression from CMV-RLuc-3′GILT in the presence of γ-IFN. This result suggests that the 3′ segment of the GILT mRNA contains a sequence that activates translation in response to IFN-γ, consistent with the above result (Figure 6).

To investigate the influence of the 5′ and 3′ regions of GILT mRNA on rapamycin-sensitive expression, HeLa cells were transfected with CMV-RLuc, CMV-5′GILT-RLuc, and CMV-RLuc-3′GILT followed by IFN-γ and/or rapamycin treatment 24 h after the transfection. Cell lysates were prepared 2 days after the treatment, and their RLuc activities were measured. In the absence of IFN-γ, rapamycin treatment significantly reduced RLuc activity in all transfected cells (Figure 7E). However, in the presence of IFN-γ, the rapamycin treatment had no effect in the CMV-RLuc-3′GILT-transfected cells but still exhibited an inhibitory effect in the CMV-5′GILT-RLuc-transfected cells. These results indicate that the 3′ segment of GILT mRNA diminishes the mTOR-dependency of translation initiation in the presence of IFN-γ.

### 2.7. STAT1 Phosphorylation Is Necessary for IFN-γ-Mediated GILT Protein Induction

The above result implies a distinct signaling pathway activated by IFN-γ that triggers GILT protein translation. To investigate whether IFN-γ-mediated GILT protein induction relies on STAT1 phosphorylation, typically induced by IFN-γ stimulation, TE671 cells were treated with IFN-γ and/or a STAT1 phosphorylation inhibitor, fludarabine (FLU) [16], for 3 days. When TE671 cells were treated with IFN-γ alone, western blotting revealed phosphorylated STAT1 and an increase in GILT proteins (Figure 8A). However, FLU treatment completely abolished STAT1 phosphorylation and IFN-γ-mediated GILT protein induction. Notably, total STAT1 levels in cells treated with FLU alone or in combination with IFN-γ were similar to untreated cells. In the absence of IFN-γ, FLU treatment reduced the basal level of GILT protein. Likewise, ddPCR analysis indicated that the basal level of IFI6 mRNA decreased with FLU treatment (Figure 8B), indicating that STAT1 is activated in the absence of IFN-γ, leading to basal translation of GILT protein and transcription of IFI6 mRNA. Taken together, these findings demonstrate that IFN-γ-mediated activation of GILT translation requires STAT1 phosphorylation.

## 3. Discussion

Our findings indicate that the induction of GILT protein by IFN-γ occurs at a translational level, whereas the expression of most target genes by IFN-γ is transcriptionally induced. GILT plays a crucial role as a host factor in both adaptive and innate immunities. Therefore, understanding the molecular mechanisms that regulate GILT expression holds significant importance in advancing our insights into adaptive and innate immunities.

Numerous lines of evidence establish a correlation between higher GILT expression levels and improved prognosis of various cancers including melanoma [17,18], lymphoma [19], glioma [20,21,22], and breast cancer [23,24]. This correlation is likely attributed to the efficient presentation of cancer antigens to T lymphocytes. Additionally, GILT’s involvement in autoimmune diseases [25,26,27] is provably linked to its impact on the generation of autoantigen peptides. Hence, our findings contribute to the comprehension of disease onset and the development of novel therapeutic strategies.

This post-transcriptional upregulation of GILT protein expression occurs in two different cell lines, HeLa and TE671 cells. This suggests that the mechanism is shared by some cell types. While HeLa cells are chronically infected with human papillomavirus, the possibility of the virus inducing the unique mechanism of GILT translation activated by IFN-γ can be ruled out, as TE671 cells were negative for the virus. However, distinct cell types may employ diverse mechanisms to regulate GILT expression. For instance, in melanoma cells, IFN-γ-mediated GILT protein induction occurs at a transcriptional level and is dependent on STAT1 [28]. Conversely, unphosphorylated STAT1 has been reported to inhibit constitutive GILT protein expression in mouse embryo fibroblasts [29,30]. Our study reveals that GILT transcription remains constitutively activated, yet its translation is suppressed in the absence of IFN-γ. Moreover, we found that treatment with the STAT1 phosphorylation inhibitor without IFN-γ further suppresses GILT translation. The unphosphorylated STAT1 may bind to the 5′ or 3′ UTR of GILT mRNA to inhibit the translation, and the phosphorylation of STAT1 induced by IFN-γ may abrogate its inhibitory effect.

The IFN-γ treatment abrogates the mTOR-dependency of GILT translation. The mTOR inhibitor, rapamycin, hampered the induction of GILT protein expression by IFN-γ. Supporting this observation, another mTOR inhibitor, everolimus, inhibits the induction of HLA class II expression by IFN-γ [31]. It is widely acknowledged that IFN-γ activates the mTOR pathway, which in turn initiates the translation of many target genes [32,33,34], whereas it was thought that mTOR was constitutively activated in HeLa cells because the CMV promoter-driven RLuc expression was suppressed by rapamycin in the absence of IFN-γ, as previously reported [15]. Interestingly, the 3′ segment of GILT mRNA rendered RLuc expression insensitive to rapamycin in the presence of IFN-γ. An unidentified host factor induced or activated by IFN-γ may initiate GILT protein translation by interacting with the 3′ segment of GILT mRNA. However, endogenous GILT expression might be dependent on mTOR in the presence of IFN-γ because the IFN-γ-mediated induction of GILT protein was inhibited by rapamycin.

This study reveals a unique mechanism of GILT expression regulation (Figure 9). Even in the absence of IFN-γ, GILT transcription remains constitutively active; however, translation is impeded by either the 5′ or 3′ segment of the full-length GILT mRNA, likely due to interactions with trans-acting inhibitory factors. When the sole UTRs were fused to the RLuc coding sequence, the expected translational regulation was not observed. Consequently, the cis-regulatory sequences overlap within the UTRs and GILT protein-coding region. Upon exposure to IFN-γ, the trans-acting inhibitory factor binding on the 3′ segment is released. A distinct translation-enhancing factor is induced by IFN-γ, which associates with the 3′ segment to initiate GILT protein translation, even though the translation inhibitor still binds to the 5′ segment.

The principal host antiviral factor induced by IFN-γ in TE671 cells is GILT, but not in HeLa cells. Specifically, GILT silencing had no impact on the antiviral activity of IFN-γ in HeLa cells, whereas, in TE671 cells, it almost competently abrogated this activity [6]. Additionally, an intriguing discrepancy was observed in the IFI6 copy number response to IFN-γ treatment between HeLa and TE671 cells. While HeLa cells exhibited a remarkable 17-fold increase, TE671 cells only displayed a modest 2- to 4-fold elevation. This finding aligns with the previous research, as IFI6 functions as a host defense factor against retroviruses [12]. In GILT-silenced HeLa cells, IFI6 inhibits retrovirus infection in response to IFN-γ treatment.

The IFN-γ signal pathway is less efficiently activated by IFN-γ treatment in TE671 cells than in HeLa cells. In the conventional IFN-γ signaling, the STAT1 protein undergoes phosphorylation by activated Jak and then is transported to the nucleus. Phosphorylated STAT1 functions as a transcription factor, leading to the increased transcription of most IFN-γ target genes within a few hours. It is most likely that the induction of IFI6 by IFN-γ also requires STAT1 phosphorylation because IFI6 induction is mediated through transcriptional activation. While the induction of GILT protein expression by IFN-γ in TE671 cells was observed to a similar extent as in HeLa cells, the induction of IFI6 was not. These results suggest that the upregulation of GILT protein expression by IFN-γ occurs through an atypical IFN-γ signal pathway, although it requires STAT1 phosphorylation. The complexity of the mechanism behind GILT protein induction by IFN-γ aligns with the observation that maximal expression of GILT protein in response to IFN-γ required an extended time period (3 days). The trans-acting factors binding to the GILT UTRs exert a significant influence on GILT protein expression, as mentioned above. Considering GILT’s involvement in various diseases including cancers and autoimmune disorders, the expression regulation by these trans-acting factors also correlates with the development of these diseases. Further study is required for the identification of the trans-acting factors.

## 4. Materials and Methods

### 4.1. Plasmids

The protein-coding region of GILT was amplified using RT-PCR (TaKaRa, Otsu, Japan) with RNA extracted from HeLa cells. The resulting PCR product was integrated into the pTargeT mammalian expression plasmid (Promega, Madison, WI, USA) [6]. Additionally, the GILT protein-coding sequence without the 5′ and 3′ untranslated regions was linked to a murine leukemia virus (MLV) vector genome expression plasmid that carries the puromycin-resistant gene [5]. Another GILT expression plasmid containing both the 5′ and 3′ untranslated regions (UTRs) (full GILT) was acquired from Origene.

A fragment containing the 5′ UTR and GILT-coding region was amplified through PCR using 5′UTR sense and GILT antisense primers (Table 1). The LA Tag DNA polymerase (TaKaRa) that has high fidelity was used in this study. Then, the resulting PCR product was integrated into the pTargeT expression plasmid (5′UTR-GILT). Similarly, a fragment containing the GILT-coding region and the 3′ UTR was obtained via PCR using GILT sense and 3′UTR antisense primers and subcloned into the pTargeT expression plasmid (GILT-3′UTR). For constructing chimeric sequences, the full-length GILT cDNA was digested with *Apa*I restriction enzyme, and the resulting 5′ half fragment was fused to the 5′ end of Renilla luciferase within the pcDNA3.1 expression plasmid (CMV-5′GILT-RLuc). Correspondingly, the 3′ half of the full-length GILT cDNA sequence was fused to the 3′ end of Renilla luciferase (CMV-RLuc-3′GILT). The construction of RLuc expression plasmids containing the UTRs of GILT cDNA proceeded as follows: The RLuc-coding sequence was amplified with PCR using 5′UTR-RLuc sense and RLuc antisense primers. The sense primer used in this process encompassed the 5′ UTR of full-length GILT cDNA as well as the 5′ portion of RLuc (Table 1). The resulting PCR product was inserted into the pcDNA3.1 vector, denoted as CMV-5′UTR-RLuc. The 3′ UTR was also amplified with PCR using 3′UTR sense and 3′UTR antisense primers and then fused to the 3′ terminus of RLuc, forming CMV-RLuc-3′UTR construct.

The promoter/enhancer region of the GILT gene was amplified with PCR using genomic DNA from HeLa cells. The nucleotide sequence was determined using the BigDye Terminator v3.1 cycle sequencing kit (Thermo Fisher Scientific, Vilnius, Lithuania). Subsequently, the PCR product was combined with the 5′ end of the nanoluciferase-coding region.

The HIV-1 Gag-Pol-Tat-Rev expression plasmid was generously provided by Dr. D. Trono [35]. The MLV Gag-Pol expression plasmid was obtained from TaKaRa. Our research group constructed the amphotropic envelope glycoprotein expression plasmid [36]. The VSV-G expression plasmid was kindly supplied by Dr. L. Chang through the AIDS Research and Reference Reagent Program, NIAID, NIH, USA [37]. Renilla, firefly, and nano luciferase expression plasmids were purchased from Promega. The expression plasmid for firefly luciferase under the control of gamma interferon activating sequence (GAS) from the LMP2 gene promoter was obtained from Dr. G.R. Stark [14].

### 4.2. Cells

Human 293T and HeLa cells were sourced from ATCC and have been under our laboratory’s care since its inception. These cell lines were cultured in Dulbecco’s modified Eagle’s medium (D-MEM) (Fujifilm-Wako, Osaka, Japan) supplemented with 8% fetal bovine serum (Capricorn Scientific, Ebsdorfergrund, Germany) and 1% penicillin-streptomycin (Sigma-Aldrich Japan, Tokyo, Japan) at 37 °C in 5% CO_2_ environment.

To generate HeLa cells stably expressing the GILT protein, the following procedure was employed: A D-MEM solution (100 μL) containing the expression plasmids for MLV Gag-Pol (1 μg), VSV-G (1 μg), and GILT-encoding MLV vector genome (1 μg) was mixed with 5 μL of the Fugene transfection reagent (Promega). The mixture was thoroughly combined and incubated at room temperature for 10 min before being added to 293T cells. After 24 h of transfection, the culture medium was replaced with a fresh medium, and the cells were cultured for an additional 24 h. Subsequently, the culture supernatant was passed through a 0.45 μm filter to eliminate cells and was then applied to HeLa cells. The HeLa cells that received the inoculum were subjected to puromycin treatment (2 ng/mL) (Wako), resulting in the selection of a pool of puromycin-resistant cells that was used for the subsequent study.

Cells were treated with 0.2 μg/mL of IFN-γ (Fuji Film Wako, Osaka, Japan), 100 μM of cycloheximide (cayman, MI, USA), 1 μM of rapamycin (Fuji Film Wako), or 2 μM of fludarabine (Selleckchem, TX, USA) as indicated.

### 4.3. Peripheral Blood Mononuclear Cells

Peripheral blood mononuclear cells (PBMCs) were isolated from a healthy donor using LSM lymphocyte separation medium (MP Biomedicals, Irvine, CA, USA). Subsequently, the isolated PBMCs were cultured in RPMI medium supplemented with 20% fetal bovine serum (Capricorn Scientific, Akita, Japan) and 1% penicillin-streptomycin (Sigma-Aldrich). The cell culture was maintained at 37 °C in a 5% CO_2_ environment. This experiment received approval from the Ethics Committee of Nagasaki University on February 20th, 2012 (Permission number: 11122884).

### 4.4. Amphotropic Murine Leukemia Virus Vector

Human 293T cells were transfected with the expression plasmids for MLV Gag-Pol, amphotropic envelope glycoprotein, and the MLV vector genome encoding Renilla luciferase, as detailed above. For HeLa cells, a pretreatment involving IFN-γ and/or rapamycin was administered for 3 days, followed by their inoculation with the culture supernatant from the transfected cells. To measure the amphotropic glycoprotein-mediated infection, Renilla luciferase activity was assessed 2 days post-inoculation utilizing a luminometer (ATTO) with the Renila luciferase activity assay kit (Promega).

### 4.5. Western Blotting

Cell lysates were prepared from an equivalent number of cells. These lysates underwent SDS polyacrylamide gel electrophoresis (Bio-Rad, Hercules, CA, USA), and the proteins were subsequently transferred onto a PVDF membrane (Millipore, Burlington, MA, USA). After blocking with 10% skim milk (Difco, MD, USA), the membrane was subjected to treatment with antibodies: goat anti-GILT antibody (T-18) (Santa Cruz Biotechnologies, TX, USA) (Figure 1A, Figure 2A and Figure 4A), mouse anti-GILT antibody (G-11) (Santa Cruz Biotechnologies) (Figure 5A, Figure 6B and Figure 8A), mouse anti-actin antibody (Santa Cruz Biotechnologies), mouse anti-phosphorylated STAT1 antibody (Cell Signaling Technology, Danvers, MA, USA), or mouse anti-total STAT1 antibody (Transduction Laboratories, Franklin Lakes, NJ, USA), followed by either HRP-conjugated protein G (Bio-Rad) or anti-mouse IgG antibody (Bio-Rad). The antibody-bound proteins were detected by the treatment of the membrane with the ECL reagents (Bio-Rad) using a chemiluminescence scanner (LI-COR Biosciences, Lincoln, NE, USA).

### 4.6. Luciferase Activity

HeLa cells were transfected with the Renilla, firefly, or nano luciferase expression plasmids or were inoculated with the amphotropic MLV vector encoding Renilla luciferase. Cell lysates were prepared using a lysis buffer of the luciferase assay kit (Promega). Renilla, firefly, or nano luciferase activity was quantified using specific substrates by a luminometer (ATTO, Tokyo, Japan).

### 4.7. Protein Stability

HeLa cells stably expressing GILT protein were treated with cycloheximide (cayman) (100 μM) for indicated periods. Cell lysates were prepared from the treated cells, and GILT protein was measured with western blotting. The intensities of mature GILT protein bands were quantitated using a chemiluminescence scanner (LI-COR Biosciences).

### 4.8. Droplet Digital PCR

Droplet digital PCR (ddPCR) was performed to measure the copy numbers of GAPDH, GILT, and IFI6 mRNA [38]. Total RNA samples were prepared from treated cells using the Trizol reagent (Thermo Fisher Scientific, Vilnius, Lithuania). Alternatively, nuclear and cytoplasm RNA samples were isolated using the RNA subcellular isolation kit (Active Motif, Carlsbad, CA, USA). Subsequently, the RNA samples were reverse-transcribed (New England BioLabs, Ipswich, MA, USA) into cDNAs using a random 9mer (TaKaRa), and these cDNAs were then combined with specific primers and the ddPCR EvaGreen Supermix (BioRad) for quantifying the copy numbers of GAPDH, GILT, and IFI6. GILT sense and GILT 532 antisense primers for GILT, GAPDH sense and GAPDH antisense primers for GAPDH, and IFI6 sense and IFI6 antisense primers for IFI6 were used in the ddPCR assay (Table 1). Droplets were generated using a droplet generator (BioRad), and the resulting samples were processed in a thermal cycler (BioRad). Finally, the fluorescence intensity of each droplet was measured using a droplet reader (BioRad) to estimate the copy number.

## 5. Conclusions

To summarize, GILT protein expression is under translational regulation in HeLa cells. Translation inhibitors bind to the 5′ and 3′ UTRs of the GILT mRNA, leading to the inhibition of GILT protein expression in the absence of γ-IFN. In response to IFN-γ, a translation enhancer that binds to the 3′ UTR of the GILT mRNA stimulates GILT protein expression. Given GILT’s relevance to multiple diseases, these trans-acting factors are also implicated in the disease processes.

## Figures and Tables

**Figure 1 ijms-25-09663-f001:**
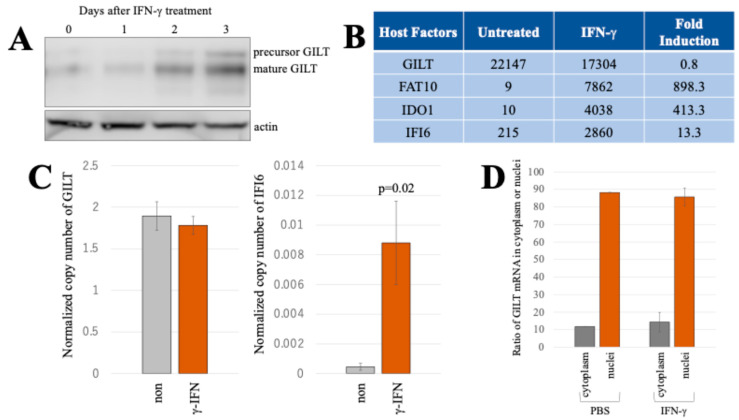
IFN-γ induces GILT protein expression but not its mRNA in HeLa cells. (**A**) HeLa cells were treated with 0.2 μg/mL of IFN-γ for indicated time period. GILT and actin proteins were analyzed using western blotting. (**B**) Fluorescent intensities of GILT, FAT10, IDO1, and IFI6 mRNA in both IFN-γ (0.2 μg/mL)-treated and untreated cells were measured with microarray (Kubo et al., 2022) [12]. Fold inductions by IFN-γ are also indicated. (**C**) The copy numbers of GAPDH, GILT, and IFI6 mRNAs were quantified using ddPCR. Normalized copy numbers are presented with error bars indicating standard deviations (*n* = 3). Significance in difference between specified groups is denoted by the *p*-value from Student’s t-test. (**D**) Copy numbers of GILT and GAPDH mRNAs in the cytoplasm and nuclei were measured using ddPCR, and the ratios of normalized copy numbers of GILT cDNAs in the cytoplasmic and nuclear fractions to the total copy numbers of GILT cDNA are indicated (*n* = 3). Error bars show standard deviations.

**Figure 2 ijms-25-09663-f002:**
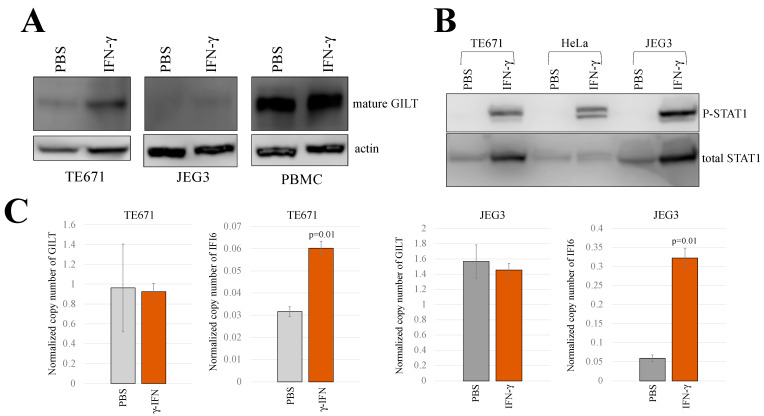
IFN-γ induces GILT protein expression but not its mRNA in TE671 cells. (**A**) TE671 cells, JEG3 cells, and PBMCs were treated with 0.2 μg/mL of IFN-γ for 3 days, and cell lysates and total RNA samples were extracted. GILT and actin proteins were analyzed using western blotting using their antibodies. (**B**) Phosphorylated and total STAT1 proteins were analyzed with western blotting, using their specific antibodies. (**C**) The copy numbers of GAPDH, GILT, and IFI6 mRNAs in TE671 and JEG3 cells were quantified using ddPCR. Normalized copy numbers are presented with error bars indicating standard deviations (*n* = 3). Significance in difference between specified groups is denoted by the *p*-value from Student’s t-test.

**Figure 3 ijms-25-09663-f003:**
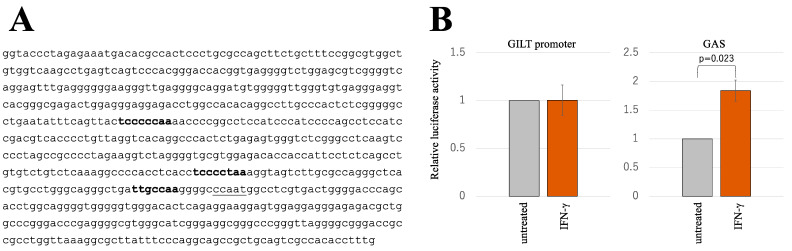
GILT promoter is not activated by IFN-γ. (**A**) The promoter/enhancer region of the GILT gene was amplified with PCR, and its nucleotide sequence is indicated. Bold and underlined letters show putative GAS and CAT sequences, respectively. (**B**) HeLa cells were transfected with expression plasmids for nano luciferase (NanoLuc) under the control of the GILT promoter/enhancer and for firefly luciferase (FLuc) under the control of the GAS sequence from the LMP2 gene and treated with IFN-γ. Cell lysates were prepared from the treated cells 3 days after the treatment. NanoLuc and FLuc activities of the cell lysates were measured (*n* = 3). Luciferase activity of the untreated cells is always set to 1. Relative luciferase activities of the IFN-γ-treated cells to those of untreated cells are indicated. Error bars show standard deviations. The *p* value between the FLuc activities in untreated and IFN-γ-treated cells is indicated.

**Figure 4 ijms-25-09663-f004:**
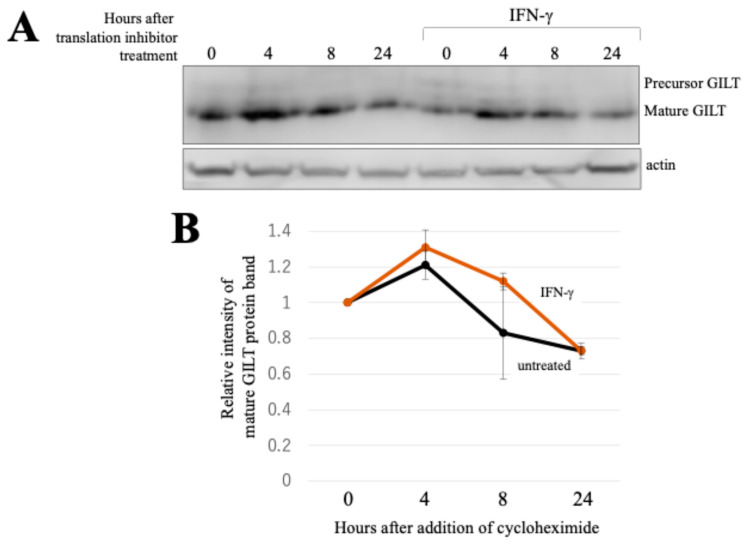
Stability of GILT protein is not changed by IFN-γ. (**A**) The translation inhibitor cycloheximide (100 μM final concentration) was added to HeLa cells transduced with an MLV vector expressing GILT and culture for indicated time periods. Cell lysates from the treated cells were analyzed with western blotting using anti-GILT or anti-actin antibody. (**B**) The intensities of the mature GILT protein detected in the western blotting analysis were measured. The GILT intensities in the untreated GILT-expressing HeLa cells are always set to 1. Relative intensities to the untreated cells are indicated (*n* = 3). Error bars indicate standard deviations.

**Figure 5 ijms-25-09663-f005:**
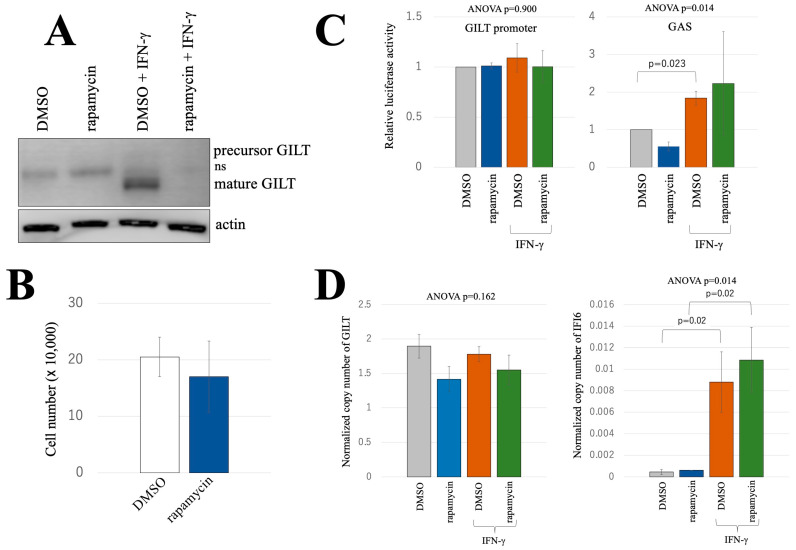
Impact of the mTOR inhibitor rapamycin on GILT protein expression, GILT promoter activity. (**A**) HeLa cells were treated with DMSO, rapamycin, and/or IFN-γ for 3 days. Cell lysates from the treated cells were analyzed with western blotting using anti-GILT and anti-actin antibodies. (**B**) HeLa cells were treated with DMSO or rapamycin for 3 days because rapamycin was dissolved with DMSO. Cell numbers were counted (*n* = 3). Error bars indicate standard deviations. (**C**) HeLa cells were transfected with the expression plasmids for NanoLuc and FLuc under the control of the GILT promoter and GAS, respectively. The transfected cells were treated with DMSO, rapamycin, and/or IFN-γ for 3 days as indicated. NanoLuc and FLuc activities were measured (*n* = 3). Luciferase activities of the DMSO-treated cells are always set to 1. Relative luciferase activities to those of the DMSO-treated cells are indicated. Error bars indicate standard deviations. The *p* values of Student’s t-test and ANOVA are shown. (**D**) The copy numbers of GAPDH, GILT, and IFI6 mRNA were measured using ddPCR. Normalized copy numbers of GILT and IFI6 are indicated (*n* = 3). Error bars show standard deviations. The *p*-values of ANOVA and Student’s t-test are indicated.

**Figure 6 ijms-25-09663-f006:**
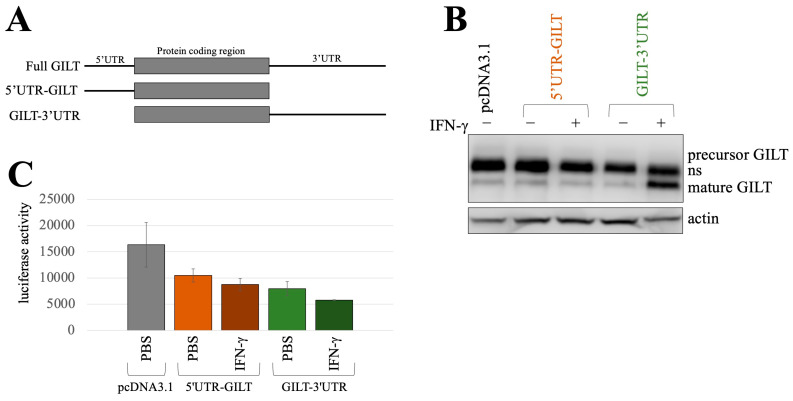
Untranslated regions of the GILT mRNA inhibit GILT protein expression. (**A**) An expression plasmid containing full-length GILT mRNA was obtained (Full GILT). A DNA fragment containing the 5′ UTR and GILT protein-coding region was amplified and ligated into pcDNA3.1 (5′UTR-GILT). A DNA fragment containing the GILT protein-coding region and 3′ UTR was amplified and ligated to pcDNA3.1 (GILT-3′UTR). (**B**) HeLa cells were transfected with the Renilla luciferase expression plasmid together with the Full GILT, 5′UTR-GILT, or GILT-3′UTR expression plasmid and then were treated with IFN-γ for 24 h. Cell lysates from the treated cells were analyzed with western blotting using anti-GILT and anti-actin antibodies. (**C**) Renilla luciferase activities of the cell lysates were measaured (*n* = 3). Error bars show standard deviations.

**Figure 7 ijms-25-09663-f007:**
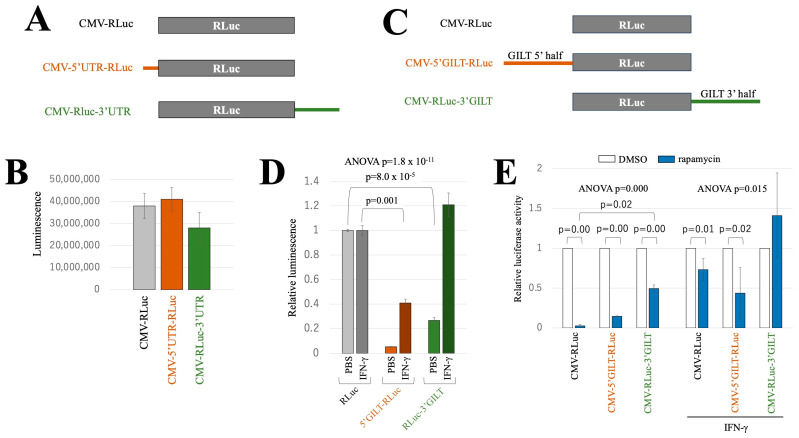
3′ untranslated region of the GILT mRNA inhibits luciferase protein expression but not in the presence of IFN-γ. (**A**) The 5′ and 3′ UTRs were fused to the 5′ and 3′ ends of the RLuc-coding region, respectively (CMV-5′UTR-RLuc and CMV-RLuc-3′UTR). (**B**) These expression plasmids were transfected into HeLa cells. RLuc activities were measured. Luminescence levels are indicated with standard deviations (*n* = 3). (**C**) The 3′ half region of the full-length GILT cDNA was linked to the 3′ end of the RLuc-coding region (CMV-RLuc-3′GILT). The 5′ half region of the GILT cDNA was fused to the 5′ end of the RLuc (CMV-5′GILT-RLuc). The resulting DNA fragments were ligated to pcDNA3.1. (**D**) HeLa cells were transfected with CMV-RLuc, CMV-RLuc-3′GILT, or CMV-5′GILT-RLuc expression plasmid and then were treated with IFN-γ for 2 days. RLuc activities of the treated cells were measured. (**E**) HeLa cells were transfected with CMV-RLuc, CMV-RLuc-3′GILT, or CMV-5′GILT-RLuc expression plasmid and then were treated with IFN-γ and/or rapamycin for 2 days. Relative luciferase activities to the CMV-RLuc-transfected cells in the absence and presence of IFN-γ are indicated (*n* = 3). Error bars show standard deviations. The *p* values of the Student’s t-test and ANOVA are demonstrated.

**Figure 8 ijms-25-09663-f008:**
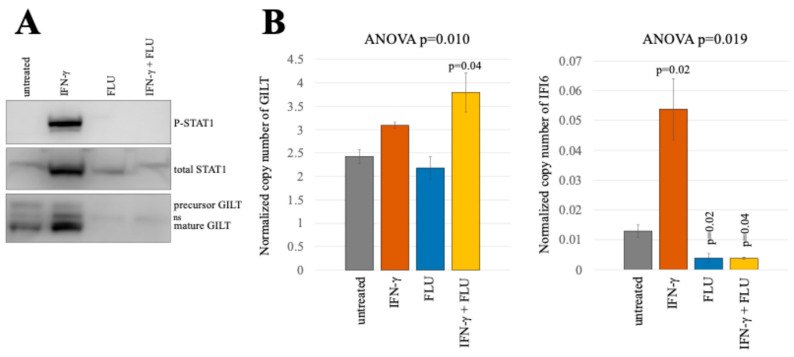
STAT1 phosphorylation is required for the initiation of GILT translation by IFN-γ. (**A**) TE671 cells were treated with IFN-γ and/or FLU, and cell lysates were prepared. Phosphorylated STAT1, total STAT1, and GILT proteins were analyzed using western blotting. (**B**) The copy numbers of GAPDH, GILT, and IFI6 mRNA were measured using ddPCR. Normalized copy numbers of GILT and IFI6 are indicated (*n* = 3). Error bars show standard deviations. The *p* values of the Student’s t-test and ANOVA are demonstrated.

**Figure 9 ijms-25-09663-f009:**
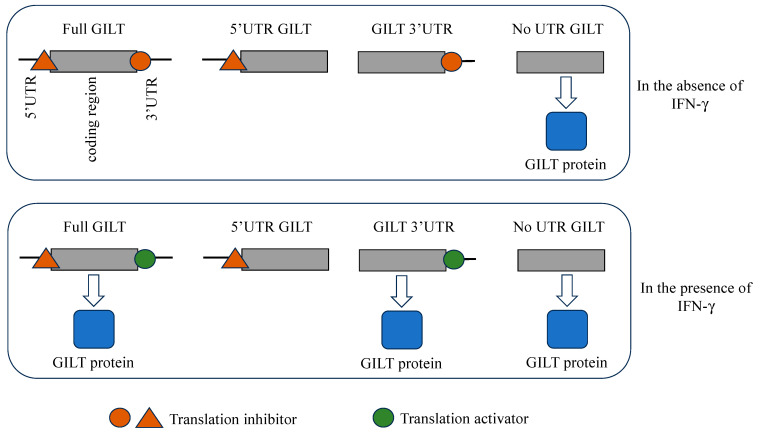
Mechanism of GILT protein expression in the absence and presence of IFN-γ.

**Table 1 ijms-25-09663-t001:** Nucleotide sequences of primers used in this study.

Primers	Nucleotide Sequences
5′ UTR sense	CTGCAGTCGCCACACCTTGC
GILT sense	ATGACTTCGAAAGTTTATGAT
GILT antisense	CACTTGAAGCAAACACTCCTG
5′ UTR-RLuc sense	CTGCAGTCGCCACACCTTTGCCCCTGCTG (5′ UTR of GILT) CG ATGACTTCGAAAGTTTATGAT (RLuc)
RLuc antisense	TTATTGTTCATTTTTGAGAACTCGC
3′ UTR sense	TGGCCGGTGAGCTGCGGAGAG
3′ UTR antisense	GCTTATTAAACTAGTTTTACTTTAGC
GAPDH sense	CCATGCCATCACTGCCACCC
GAPDH antisense	GCCAGTGAGCTTCCCGTTCAG
GILT 532 antisense	CTACAGGCATAGTGGCAGACT
IFI6 sense	GCGCGCGGCGCCACCATGCGG
IFI6 antisense	TGGCTACTCCTCATCCTCCTC

## Data Availability

All data are shown in this paper.

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
