# Peer review of "Post-Transcriptional Induction of the Antiviral Host Factor GILT/IFI30 by Interferon Gamma"

_ijms, 2024, doi:10.3390/ijms25179663_

Round 1

Reviewer 1 Report (Previous Reviewer 1)

Comments and Suggestions for Authors

no further comments

Author Response

No comment

Reviewer 2 Report (New Reviewer)

Comments and Suggestions for Authors

The work from Nakamura and Kubo is based on previous observations and sought to address how IFN-γ treatment induces GILT expression without upregulating the mRNA level of GILT. Their findings described an underlying mechanism involving an intrinsic inhibitory element within the UTRs of GILT mRNA, which can be antagonized upon IFN-γ treatment. While this reviewer acknowledges the potential interest to readers, the evidence presented is insufficient to support the authors' conclusions. Better-controlled and more rigorously designed experiments are necessary to strengthen this manuscript.

  1. The entire work described in this manuscript relied on the treatment of IFN-γ, with validation showing only a few-fold increase in IF16 transcript level. Instead of this limited induction, why did the authors not use FAT10 or IDO1 as indicators, both of which showed robust responses with hundreds of folds induction in Figure 1B?
  2. The pattern of GILT blotting changed from figure to figure. For example, the blotting of GILT in HeLa cells shows both precursor and mature GILT in untreated HeLa cells in Figure 1A, but only precursor GILT in Figure 5A. Did the authors validate the antibody with ectopically expressed GILT to confirm correct recognition of both precursor and mature GILT?
  3. What was the loading control for Figure 4A?
  4. Figure 6 is completely negative and not informative. It should be excluded from the manuscript.
  5. There is no data confirming that the mRNA transcribed from the constructs in Figure 7A was equal or at least comparable. Without this, it is hard to evaluate the effects of either UTR on GILT translation.
  6. It is interesting to observe that the 3’UTR affects the translation of mRNA, while most previous works indicate that the 3’UTR plays an essential role in the stability and localization of mRNA. Did the authors examine whether the deletion of GILT 3’UTR results in alterations in mRNA localization (nucleus vs. cytosol) and/or loading rate to ribosomes? Or at least please discuss about the possibility.
Comments on the Quality of English Language

Some wording should be more accurate, for example in line 58, what does professional cells mean? 

Author Response

Comment 1: The entire work described in this manuscript relied on the treatment of IFN-gamma, with validation showing only a few-fold increase in IFI6 transcript level. Instead of this limited induction, why did the authors not use FAT10 or IDO1 as indicators, both of which showed robust responses with hundreds of folds induction in Figure 1B?

Responses 1: Thank you for your valuable comments. As the reviewer pointed out, the mRNA levels of FAT10 and IDO1 were significantly more elevated by IFN-gamma compared to IFI6. Currently, we are investigating the molecular mechanism by which IFI6 inhibits retrovirus infection, and we have developed a protocol to measure IFI6 mRNA level using ddPCR. Although the increase in IFI6 mRNA levels induced by IFN-gamma was less pronounced than that of FAT10 and IDO1, it is indeed clear that IFN-gamma caused a noticeable elevation in IFI6 mRNA levels. Therefore, these is no reason to consider IFI6 unsuitable as an indicator of IFN-gamma signal activation. We hope the reviewer can understand our response.

Comment 2: The pattern of GILT blotting changed from figure to figure. For example, the blotting of GILT in HeLa cells shows both precursor and mature GILT in untreated HeLa cells in Figure 1A, but only precursor GILT in Figure 5A. Did the authors validate the antibody with ectopically expressed GILT to confirm correct recognition of both precursor and mature GILT?

Response 2: Initially, we used a goat anti-GILT antibody (T-18) (Santa Cruz Biotechnologies), which specifically detected only precursor and mature forms of GILT (Figures 1A, 2A, and 4A). Unfortunately, we exhausted this antibody during the study, and attempted to reorder it. However, the antibody was unavailable, so we purchased a different antibody, a mouse anti-GILT antibody (G-11) (Santa Cruz Biotechnologies). The new antibody detected an additional nonspecific band between the precursor and mature GILT proteins as indicated in (Figures 5A, 6B, and 8A).

Mature GILT was detected in Figure 6B, but it was not visible in Figure 5A. This discrepancy is likely due to the longer exposure time used in Figure 6B, which becomes apparent when comparing the intensities of the nonspecific bands in Figures 5A and 6B.

Comment 3: What was the loading control for Figure 4A?

Response 3: As the reviewer advised, we performed western blotting using an anti-actin antibody.

Comment 4: Figure 6 is completely negative and not informative. It should be excluded from the manuscript.

Response 4: As the reviewer suggested, we have deleted this figure and the corresponding sentences.

Comment 5: There is no data confirming that the mRNA transcribed from the constructs in Figure 7A was equal or at least comparable. Without this, it is hard to evaluate the effects of either UTR on GILT translation.

Response 5: To evaluate transfection efficiency, HeLa cells were transfected with the indicated GILT expression plasmids along with a luciferase expression plasmid. The description of this experiment has been revised as follow:

“These plasmids were introduced into HeLa cells together with a luciferase expression plasmid in which the expression is under the control of MLV LTR, followed by treatment with IFN-gamma for 1 day, because endogenous GILT protein was not detected by IFN-gamma treatment for 1 day (Figure 1A). Cell lysates were prepared from the treated cells. GILT protein levels were assessed via western blotting, and their luciferase activities were measured to estimate the transfection efficiency.” (line 238-244, page 7).

“However, upon treatment of HeLa cells transfected with the GILT-3’UTR construct with IFN-gamma, GILT protein expression was evident,although the luciferase activity in those cells was comparable to the luciferase activity of other cells.” (line 262-265, page 8).

Comment 6: It is interesting to observe that the 3'UTR affects the translation of mRNA, while most previous works indicate that the 3'UTR plays an essential role in the stability and localization of mRNA. Did the authors examine whether the deletion of GILT 3'UTR results in alterations in mRNA localization (nucleus vs. cytosol) and/or loading rate to ribosomes? Or at least please discuss about the possibility.

Response 6: As the reviewer advised, we analyzed the cellular localization of GILT mRNA in both untreated and IFN-gamma-treated cells. However, no changes was observed following IFN-gamma treatment (Figure 1D). The following sentences were added:

              “Transcription and translation occur in the nuclei and cytoplasm, respectively. Therefore, the transport of GILT mRNA from the nuclei to the cytoplasm may be inhibited in untreated HeLa cells to suppress GILT protein expression. To test this hypothesis, RNA samples were isolated from the cytoplasmic and nuclear fractions of untreated and IFN-gamma-treated HeLa cells. The copy numbers of GILT and GAPDH cDNAs were quantified using ddPCR, and the normalized copy numbers of GILT cDNA in the cytoplasm and nuclei were compared. The localization of GILT mRNA was unaffected by IFN-gamma treatment (Figure 1D). These findings suggest that GILT protein expression in untreated cells is inhibited by a mechanism other than the suppression of GILT mRNA transport to the cytoplasm.” (line 90-99, page 2-3).

This manuscript is a resubmission of an earlier submission. The following is a list of the peer review reports and author responses from that submission.

Round 1

Reviewer 1 Report

Comments and Suggestions for Authors

This is a very interesting paper that describes a new pathway for IFN-g biological activity, demonstrating a new effect on protein translation.  The manuscript is well written and the data of interest but there are a few issues.

1. The authors should change g-IFN to IFN-g throughout the manuscript

2. The RNA analysis is performed after 3 days of IFN treatment  I am concerned that the authors might have missed the peak of RNA expression as IFN-g can induced gene expression as early as 3-4 hours after treatment.  Have the authors looked at much earlier timepoints?

3. Are the effects on translation STAT1 dependent?   As this is a newly defined consequence of IFN-g treatment, it might not be, instead it might be NfKB dependent.

4.  Do the JEG3 cells happen to be making Type 1 IFN?  In the absence of treatment are they expressing phosphorylation STAT1 or STAT3?  Additionally, have the authors looked at pSTAT3 as STAT3 could form a heterodimer with STAT1.

Reviewer 2 Report

Comments and Suggestions for Authors

The authors describe the post-transcriptional regulation of GILT (gamma-interferon-inducible lysosomal thiol reductase) protein expression by gamma interferon and aim to elucidate the precise molecular mechanisms by which gamma interferon induces GILT protein expression. They observed that GILT mRNA is constitutively transcribed in the absence of gamma interferon, but the translation process is hindered by both the 5’ and 3’ segments. However, several comments should be considered by the authors.

Although the investigators showed that gamma interferon induced GILT protein expression in TE671 cells (Fig 3A), the results are insufficient to support the specificity of the indicated upregulation. How about other similar cell lines?

Versatile mTOR is closely associated with protein translation, so it is understandable that rapamycin could inhibit GILT protein expression even in the presence of gamma interferon (Fig 6). Additionally, the clarity of the mature GILT band could be improved (Fig 6A).

The untranslated regions of GILT mRNA have been shown to inhibit GILT protein expression (Fig 8), but the relationship with gamma interferon treatment requires more evidence to bridge the significant gap.

Reviewer 3 Report

Comments and Suggestions for Authors

The experimental strategy they followed to describe the antiviral mechanism involving the GILT protein and IFN in two cell lines HeLa and TE671 is interesting, although there are certain differences between them. You mention that the inhibition mechanism can be transcriptional or translational depending on the cell type. I ask, could the translational inhibition in HeLa be influenced because this cell line is infected with papillomavirus? Can these mechanisms you describe be modified depending on the virus that infects the cell?